# Comparative Analysis of Patient Satisfaction Surveys—A Crucial Role in Raising the Standard of Healthcare Services

**DOI:** 10.3390/healthcare11212878

**Published:** 2023-11-01

**Authors:** Karoly Bancsik, Codrin Dan Nicolae Ilea, Mădălina Diana Daina, Raluca Bancsik, Corina Lacramioara Șuteu, Simona Daciana Bîrsan, Felicia Manole, Lucia Georgeta Daina

**Affiliations:** 1Faculty of Medicine and Pharmacy, Doctoral School of Biomedical Sciences, University of Oradea, 1 December Sq., 410081 Oradea, Romania; 2Faculty of Medicine and Pharmacy, University of Oradea, 1 December Sq., 410081 Oradea, Romania; 3Clinical Emergency Hospital “Avram Iancu”, 410027 Oradea, Romania; 4Department of Psycho-Neurosciences and Recovery, Faculty of Medicine and Pharmacy, University of Oradea, 1 December Sq., 410081 Oradea, Romania; 5Department of Surgical Disciplines, Faculty of Medicine and Pharmacy, University of Oradea, 410081 Oradea, Romania

**Keywords:** quality management, patient-perceived quality assessment (PPQA) questionnaire, quality of medical care, satisfaction, general impression of the hospital

## Abstract

(1) Background: The study aimed to assess the patients’ perception of the quality of the medical staff’s care, the hotel’s services, and the hospital’s overall impression as well as to determine the best rating scale through a comparative analysis of patient satisfaction questionnaires. (2) Methods: A retrospective study was performed based on satisfaction questionnaires addressed to the patients hospitalized in the Orthopedics and Traumatology departments of the County Clinical Emergency Hospital Oradea between 2015 and 2019. Three different types of questionnaires were used during the study period, with the number of questions varying between 30 (variant A) and 37 (variant C). The evaluation was done using the Likert scales with three, four, or five answer variables. (3) Results: The items that we found to be present in all three categories of surveys and for which at least two different questionnaire variants used the Likert scales with various answer variables were chosen. In terms of the treatment given by the medical staff, hotel services, and the overall perception of the hospital, the patients perceive a higher level of quality. (4) Conclusions: The level of patient overall satisfaction or general impression about the hospital is strongly dependent on the quality of medical care provided by the doctors and the specific hotel conditions of the hospital. The quality assessment using the Likert rating scale with five binary variables is more accurate.

## 1. Introduction

The World Health Organization emphasizes that in the health system, the provision of quality medical services is essential. The actions of the health system must be responsive to the needs of the population, treating people with respect [1]. In the hospital, these needs of the population can be assessed by evaluating the patients’ satisfaction, an aspect regulated in Romania by related legislation and the recommendations of the National Health Quality Management Authority (ANMCS). At the national level, public hospitals have adopted a patient feedback process that assesses patient satisfaction in terms of the standard of services, respect of patient rights, and ethical conduct of medical and sanitary staff [2]. As part of the ANMCS certification process, hospitals are required to provide patient satisfaction surveys [3]. Since each hospital is free to apply its own questionnaire in accordance with the ANMCS standards, there is no universal patient satisfaction survey model to be used nationally. “The patient at the center of concerns” and “patient satisfaction” are established as two fundamental concepts of quality in health by ANMCS in the National Strategy for Quality Assurance in the Health System published in 2018. ANMCS also suggests a national portal for the centralized collection of feedback and patient satisfaction levels [4].

The implementation of a unitary/standardized monitoring mechanism of the performance in health facilities, in order to reduce practice variability, constitutes an important component of the processes in improving the health system [5,6]. Currently, medical institutions are motivated to organize their quality management structures, prevent medical risks, and adjust current practices to the standards of evidence-based medicine [7,8]. The quality of health services, unlike other tangible goods, cannot be assessed until after it has been provided. The patients’ reaction is subjective, as it is based on expectations and perceptions that may vary from person to person or even the same person at different times [9,10]. A systematic evaluation of perceived quality requires tools capable of monitoring the expectations of all patients [11].

Quality assessment represents the systematic measurement of the current level of quality achieved by a unit or a system and consists of quantifying the level of performance according to the provided standards [12,13]. The first step in the quality assessment process is to identify areas for improvement [14,15]. By adopting standards with the support of all parties involved, quality management in health ensures that the patient is at the center of concern while the quality and safety of medical assistance are continuously improved. It also ensures that the best operational and managerial procedures are used. The achievement of this mission aims at improving the organizational framework and changing the culture of health organizations, through actions that promote the concept of quality in health [4].

Patient-perceived quality assessment (PPQA) has become a critical component in the development and improvement of health services and patient care. This is based on the assessment of medical services by the patient, and the main purpose is to develop and implement measures that improve the quality of medical services and the experiences of the consumers of medical services [16,17]. In recent years, the associations that represent the interests of the patients and the patients have become more and more involved in contributing and increasing quality in the medical field [18,19]. Considering the fact that in the Oradea County Emergency Clinical Hospital, there are specialized committees that actively supervise the daily activities while they are being carried out, they can modify, supplement, or adopt corrective or preventive measures when necessary [20].

Assessing the patient-perceived quality regarding medical care is essential for maintaining a high level of quality within the organization [21,22]. An increasing or decreasing trend in perceived quality may indicate effective/ineffective quality management or inaccurate assessment methodology [19,23]. Thus, we propose the analysis of the three variants of satisfaction questionnaires following the correlation between the perceived quality, the variant, and the scale used. The quality assessment using the Likert scale with five numeric response values was used in questionnaire variant C for the quality evaluation of medical care.

The present study aimed to evaluate the quality perceived by the patient in the hospital and to identify optimal evaluation scales that lead to the improvement of the quality of medical services. The implementation of a system with increased assessment accuracy will help both in decision-making by the hospital management and in the development of a sustainable management strategy. To achieve the proposed goal, the following objectives were defined:Comparative analysis of the variants of satisfaction questionnaires applied in the studied period. Identifying an optimal rating scale by benchmarking patient satisfaction questionnaires and evaluation of the quality perceived by the patient regarding the care provided by the medical staff, hotel services, and the general impression of the hospital.Evaluation of the patient’s overall opinion (general impression) in relation to the quality of medical care provided by the hospital staff and hotel conditions in the hospital.

## 2. Materials and Methods

### 2.1. Study Design

The study was carried out in the County Clinical Emergency Hospital Oradea (CCEHO) by analyzing patient satisfaction questionnaires. The CCEHO is a tertiary level public hospital located in N–W Romania, which provides medical assistance for approximately 200,000 inhabitants of the Municipality of Oradea and emergency medical services for a territorial population of approximately 600,000 inhabitants.

The study period selected was 2015–2019. The years 2020–2022 were not included in the analysis due to the following considerations: admission restrictions and exceptional measures due to the COVID-19 pandemic (2020–2021) and the significant structural changes in the hospital in 2022. It should be noted that until the end of 2022, patient satisfaction questionnaires were applied in the hospital, the content of which remains unchanged since 2019 (variant C of the questionnaire presented in the study).

The number of beds decreased from 885 beds (2015–2017) to 861 beds (2018–2019) over the studied period. The average number of patients discharged annually was around 40,000, and the number of applied questionnaires was around 3500/year, which represents an average of about 8.75% of patients discharged annually who fully responded to the administered questionnaires.

The Orthopedics and Traumatology wards, which have a similar number of beds (33 beds in Orthopedics and Traumatology Ward 1 and 30 beds in Orthopedics and Traumatology Ward 2) and offer the same kinds of medical services, were chosen as the study’s sample wards. The average discharged patients from each department analyzed varied between 1150 and 1300/year.

Ten reports, based on 1215 questionnaires, during the studied period were evaluated. A number of 1101 questionnaires were valid, the exclusion criteria being incomplete questionnaires (the patients did not respond on all questions, *n* = 114). The reports were based on three versions of the satisfaction questionnaire; version A was used in the period 2015–2016, version B in the period 2017–2018, and version C in 2019, the latter having a set of revised questions according to the necessary standards for the hospital accreditation. Evaluation of patient satisfaction is part of the implementation of a unitary/standardized monitoring mechanism for the performance in health facilities, in order to reduce practice variability, focus on quality, and provide feedback for quality improvement [4].

### 2.2. Data Collection and Tools

The three types of questionnaires contained sets of 30, 31, and 37 standardized questions. They were elaborated under the monitoring obligations of patient satisfaction resulting from the Framework Agreement regarding the conditions for the provision of medical assistance in the Romanian health care system [24]. The questionnaires include 7 sections/domains with 1–15 questions per domain. These domains were demographic data, accessibility/admission, hotel conditions, quality of medical care, patient safety and rights, overall satisfaction, and finally observations/suggestions. The survey variants are presented in Table 1.

In order to measure patient satisfaction, Likert scales with 3, 4, or 5 answer variables were used, depending on the three versions of the questionnaire. Patients responded by using common rating scales—either by selecting a response (unsatisfactory, good, or very good) or rating between 1 and 5 (Very good) [25,26,27]. The questionnaire types and the correspondence of scales are presented in Table 2.

To achieve the proposed objectives, we analyzed 2 domains: quality of medical care and overall satisfaction. The questions that we find in all three types of questionnaires and in which Likert scales with different answer variables are used in at least two questionnaire variants were selected. The selected questions are:
Q15A, Q15B: Are you satisfied with the hotel conditions (accommodation)?□ Very satisfied □ Satisfied □ UnsatisfiedQ14C: Are you satisfied with the accommodation conditions—salon (equipment, facilities)?□ 1 □ 2 □ 3 □ 4 □ 5 □ I do not answerQ17A, Q18B: The quality of medical care provided by:Your doctor: □ Unsatisfactory □ Good □ Very goodNurses: □ Unsatisfactory □ Good □ Very goodCaregivers: □ Unsatisfactory □ Good □ Very goodQ15C: The quality of medical care provided by (1—unsatisfactory.... 5—very good):Your doctor: □ 1 □ 2 □ 3 □ 4 □ 5 □ I do not answerNurses: □ 1 □ 2 □ 3 □ 4 □ 5 □ I do not answerCaregivers: □ 1 □ 2 □ 3 □ 4 □ 5 □ I do not answerQ29A, Q30B: Your general impression of the hospital?□ Very satisfied □ Satisfied □ UnsatisfiedQ36C: Your general impression of the hospital?□ 1 □ 2 □ 3 □ 4 □ 5 □ I do not answer

### 2.3. Statistical Analysis

For general characteristics of the study population, the chi-square test and t-test were performed to compare the differences between groups, with *p*-value < 0.05 being considered significant. To determine the association between the questionnaire type and patient satisfaction, we applied linear regression analyses. The results were considered significant at a *p*-value lower than 0.05. We used “very good”, “good”, and rating 3, 4, 5 as the outcome for the logistic regression and the other responses were grouped into one group [28,29].

For the evaluation of the Likert scale internal consistency, we applied Cronbach’s alpha tests. For data centralization and statistical analysis, Microsoft Office programs were used.

### 2.4. Participants

Of the 1215 questionnaires applied during the analyzed period, 90.6% (*n* = 1101) questionnaires were validated, and 47.2% (*n* = 520) from the Orthopedics 1 (O1) department and 52.8% (*n* = 581) from the Orthopedics 2 (O2) department, respectively, presented valid questionnaires. Most patients were from rural areas (54.1%, *n* = 524), and 52.2% (*n* = 579) were women. Of the 1101 responding patients, 88% (*n* = 969) provided complete information about their domicile and 97.1% (*n* = 1069) provided complete information about their level of education. From the sample of patients who provided answers regarding education, 56.4% (*n* = 603) had a high school diploma or a higher degree of education, 33.4% (*n* = 357) had 8th class/grade, and 10.2% (*n* = 109) had primary education. Full sample characteristics are shown in Table 3.

### 2.5. Ethics

To carry out the study, consent regarding access to the database was initially re-quested and obtained; later, data were retrieved and processed. The data were provided by the Statistical Service of County Clinical Emergency Hospital Oradea. This study used secondary data without the patient’s information and the study was approved by the Ethical Committee of County Clinical Emergency Hospital Oradea, Romania with no. 459/08.01.2019.

## 3. Results

To evaluate the quality of medical care provided by the attending physician, nurses, and caregivers, the ordinal Likert scale with three answer values was used for versions A and B; the Likert scale with five numerical answer values was used for version C. On average, 98.8% of patients rated the quality of medical care provided by the attending physician as good and very good in the Orthopedics 1 department, and 98.2% in the Orthopedics 2 department reported similarly during the studied period. Between 2015 and 2017, a high degree of quality was maintained (100%) in both departments, followed by a sustained decrease until 2019, with almost 8% less than in 2017, the data being presented in Figure 1a. Quality assessment using the ordinal Likert scale with three response values did not reveal major changes regarding perceived quality, and the maximum difference between the years was 0.85%. Comparatively, the use of the five-point Likert scale in 2019 revealed a difference of 7.1% compared to 2018 (*p*-value < 0.001). A percentage of 2.5% refused to answer this question. We applied the logistic regression model to assess the quality of medical care provided by the attending physician in relation with the patient’s overall opinion (general impression), which is presented in Figure 1b,c, and we found that there is a direct relationship between these two variables (*p*-value < 0.001).

The evaluation of the quality of medical care attributed to the nurses revealed that 82.5% of patients considered the quality to be good and very good in the Orthopedics 1 department, compared to 80.9% in the Orthopedics 2 department on average in the studied period. Following Figure 2, an upward trend can be observed regarding the quality of medical care provided by nurses from 2015 to 2019, increasing from 81.7% to 83.5% in the Orthopedics 1 ward, compared to an increase from 79.3% to 82.8% in Orthopedics Section 2. The quality assessment using the ordinal Likert scale with three response values and the Likert scale with five numerical values did not reveal major changes regarding the perceived quality, and the maximum difference between the years was 4.2% (*p*-value < 0.001). A percentage of 2.5% refused to answer this question. We applied the logistic regression model to assess the quality of medical care provided by the nurses in relation with the patient’s overall opinion (general impression), which is presented in Figure 2, and we found that there is a weak relationship between these two variables (*p*-value = 0.05).

The evaluation of the quality of medical care attributed to caregivers revealed that 79.1% of patients consider the quality to be good and very good in the Orthopedics 1 section, compared to 80.9% in the Orthopedics 2 section on average in the studied period. In Figure 3, an 8% decrease in the quality of medical care provided by nurses in the Orthopedics Department 2 can be observed from 2015 to 2019. The quality assessment using the ordinal Likert scale with three response values compared to the Likert scale with five numerical values did not reveal major changes regarding the perceived quality (*p*-value < 0.001). A percentage of 2.5% refused to answer this question. We applied the logistic regression model to assess the quality of medical care provided by the caregivers in relation with the patient’s overall opinion (general impression), which is presented in Figure 3, and we found that there is a weak relationship between these two variables (*p*-value = 0.01).

On average, 95.3% of patients rated the hotel conditions as good and very good in the Orthopedics 1 ward, compared to 94.8% in the Orthopedics 2 department during the studied period. In Figure 4a, a decrease in the quality of hotel conditions in the Orthopedics 1 department from 2015 to 2019 can be seen, falling from 99.1% to 91.3%. Quality assessment using the ordinal Likert scale with three response values did not reveal relative changes regarding the perceived quality, and the maximum difference between the years was 7.4%. Comparatively, the use of the five-point Likert scale in 2019 revealed a difference of 4.81% compared to 2018 (*p*-value < 0.0001). We applied the logistic regression model to assess the quality of hotel conditions in the hospital in relation with the patient’s overall opinion (general impression), which is presented in Figure 5, and we found that there is a strong relationship between these two variables (*p*-value < 0.001).

On average, 95.8% of patients had a good and very good overall impression of the hospital during the period studied. A 12% decrease in positive responses regarding the overall impression of the hospital can be seen from 2015 to 2019. Quality assessment using the three-point ordinal Likert scale did not reveal relative changes in the perceived quality, and the maximum difference between the years was 6.7%. Comparatively, the use of the five-point Likert scale in 2019 revealed a difference of 17.2% compared to 2018 (*p*-value < 0.001).

The evaluation of the Likert scale internal consistency shows that using the Type C questionnaire variant provides a reliable tool for quality assessment. The results are presented in Table 4.

## 4. Discussion

In recent years, the need to implement a new quality management strategy has become increasingly important in the current economic, legislative, and social context [30,31]. The pressure exerted by the population on the medical staff, the medical service provider, and the government requires the urgent development of these strategies. Initially, to develop a new strategy that has a sustainable character, it is necessary to implement a system for evaluating the perceived quality as accurately as possible, which measures quality in real-time [32,33]. The evaluation of the quality of medical services perceived by the patients is carried out through satisfaction questionnaires [34,35] and represents a priority issue both at the level of primary medical care and at the level of the hospital [36,37]. A systematic analysis of the specialized literature published by Derriennic J and collaborators in 2022, regarding patient self-assessment tools on the quality of primary medical care in multi-professional clinics, reveals that the measurement properties of these tools are weak and development and validation of a generic tool is necessary [38]. At the hospital level, the quality of medical services from the perspective of patient satisfaction analysis remains a continuous concern [39,40], representing a useful tool for the manager in performance evaluation [41]. Managers should not only focus on improving service quality, but also on overall strategic planning [42].

Personal characteristics along with patients’ expectations, their health status, as well as the characteristics of the health system influence patient satisfaction or dissatisfaction [43]. Patient expectations vary from one socio-demographic group to another; therefore, in order to remove the influences given by these characteristics, each population group responding to the three types of questionnaires was analyzed, resulting in homogeneity of the group. The evaluation of the quality of medical care provided by the doctor and nurse is essential in obtaining patient satisfaction, and similar analyses are being reported in other published studies [44,45,46]. This analysis looks at the medical or surgical treatment provided by doctors, the clarity in communication with patients, the provision of clear explanations of what they were doing, and the compassion, care, and empathy of nurses. Patient satisfaction increased during the analyzed period and no significant differences were obtained on the three types of questionnaires regarding the medical assistance provided by the on-call staff, attending physicians, and nurses. A study carried out in the context of the epidemic of COVID-19 and published in 2021 by Nguyen NM shows that if the hospital services provided by the medical staff (service attitude and professional capacity of the medical team) are good, then the tranquility and satisfaction of the patients will be increased [47]. Also, improving communication skills can lead to better patient satisfaction and positively influence health outcomes [48]. The doctor’s expertise is much more important for patients than satisfaction with nurses or other staff; similar results were obtained by Gavurova B et al. in 2021 [49].

Hospital hotel conditions refer to accommodation (lounge, equipment, furniture, linen, cleanliness, hospital effects, facilities, temperature), cleanliness, and food; experts consider that these factors include elements related to the economy, culture, security, public welfare services, clinical care services, patient referral, and staff identification [50]. In the study conducted, the degree of patient satisfaction with hotel conditions is high, with a slight decrease in 2019 compared to previous years. Although there were no essential interventions on hotel conditions, a possible explanation for the decrease in patient satisfaction could be the improvement of the assessment methodology, with an emphasis on cleanliness and food, where lower scores were obtained.

Reliability, security, and patient safety issues are all included in the examination of the hospital’s overall image. Healthcare professionals need to be aware of the fact that today’s patients are more informed and quality-conscious than ever before because they deal with people’s lives and health on a daily basis. A similar study conducted by Al Awadh M in 2022 in Saudi Arabia, using the multi-criteria decision-making technique, provides the management with valuable information about the factors that demonstrate how satisfied patients are in the hospital. Therefore, hospitals have the ability to improve the quality of their services and offer patients and clients an even better level of satisfaction by addressing the unique limits they encounter, assisting patients in making more informed decisions [51].

Due to the technological advances of the 21st century, the multidimensional aspects that the Internet offers must also be taken into account, more specifically the interaction that society has through social networks. A 2021 study carried out by A Rahim AI in Malaysia demonstrates that tracking Facebook reviews using machine learning techniques offers useful real-time data that are not accessible through traditional quality measurements or surveys. This study found that patients in Malaysia were generally happy with the care they received from public hospitals. All SERVQUAL (a multi-dimensional research instrument meant to capture consumer expectations and perceptions of service) characteristics were strongly associated with a favorable feeling, with the exception of the physical ones. The authors suggest that hospital administrators and decision-makers use this special stream of data to better understand patients’ healthcare experiences and the standard of treatment they receive, even though many hospitals have their own Facebook sites and actively monitor them [52].

The relationship between resource allocation and patient satisfaction level should not be overlooked in the ongoing improvement of quality in the hospital. According to a study done by Valls Martinez MDC in 2021, the amount of money spent directly affects patient happiness and, consequently, the standard of the healthcare system. Spending on primary care should be increased, particularly for specialized medical care and diagnostic tools. Additionally, cutting back on drug use in favor of complementary therapy is viewed favorably. Similarly, expenses have an impact on available resources and these, in turn, have a positive influence on the level of use and a negative impact on mortality [53].

The study carried out allows the comparative analysis of the variants of patient satisfaction questionnaires used in the hospital. The important role of patient satisfaction measurement tools that must be used for interventions to improve the quality of hospital health services is also emphasized in the specialized literature [19,54,55]. The implementation of a standardized questionnaire at the national level allows the unitary assessment of the quality of medical assistance and services provided by medical units, representing a reliable tool for improving the health status of the population [56].

### Limitations of the Study

The aim of the study was not to make comparisons between departments, but to assess the patients’ perception of the quality of the medical staff’s care, the hotel’s services, and the hospital’s overall impression as well as to determine the best rating scale through a comparative analysis of patient satisfaction questionnaires.

The statistical differences in the demographic data between the two wards did not influence their results.

The data available for research do not cover all patients admitted and discharged between the studied period.

## 5. Conclusions

In recent years, the need for quality management improvement has become increasingly important in the current economic, legislative, and social context. The present study confirms the hypothesis that the level of patient overall satisfaction or general impression about the hospital is strongly dependent on the quality of medical care provided by the doctors and the specific hotel conditions of the hospital. We found that the medical care provided by the nurses and caregivers has low impact on the general impression. This study highlights that the level of quality of medical care provided by doctors, nurses, and caregivers as well as the specific hotel conditions of the hospital are important factors in terms of patient satisfaction, and the results of this study highlighted the need to improve some of these conditions. The implementation of a unitary/standardized monitoring mechanism of the performance in the Orthopedics and Traumatology departments revealed that the use of the five-point Likert scale provides an accurate assessment of perceived quality. For further improvement and complexity enhancement, the authors of this study suggest that the questions in the patient satisfaction questionnaire be formulated in such a way as to allow the use of the Likert rating scale with five binary variables.

## Figures and Tables

**Figure 1 healthcare-11-02878-f001:**
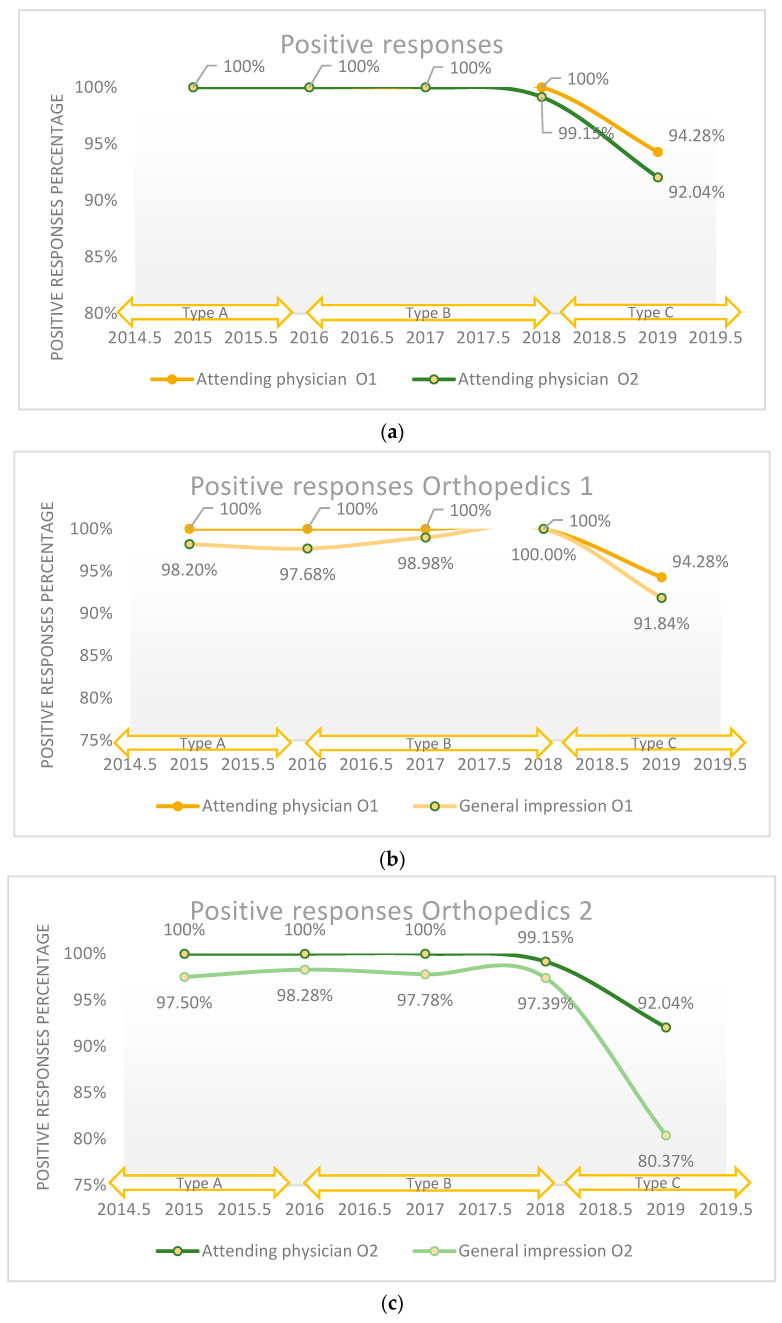
(**a**) The positive responses regarding the quality of the medical care provided by the attending physician reported annually; (**b**) The positive responses in Orthopedics 1 department regarding the quality of the medical care provided by the attending physician in relation with patient’s overall opinion (general impression) reported annually; (**c**) The positive responses in Orthopedics 2 department regarding the quality of the medical care provided by the attending physician in relation with patient’s overall opinion (general impression) reported annually.

**Figure 2 healthcare-11-02878-f002:**
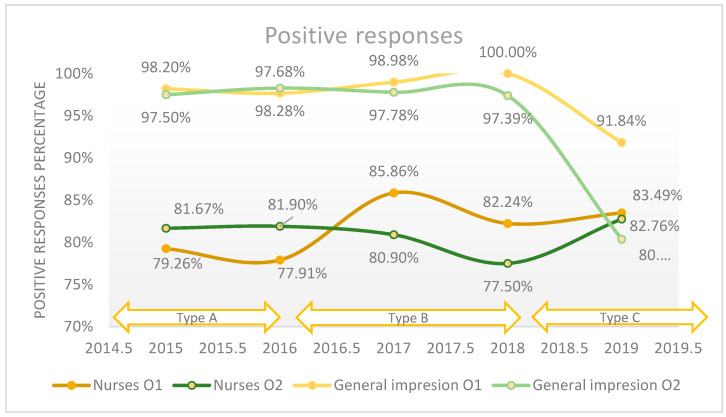
The positive responses regarding the quality of the medical care provided by nurses in relation with patient’s overall opinion (general impression) reported annually.

**Figure 3 healthcare-11-02878-f003:**
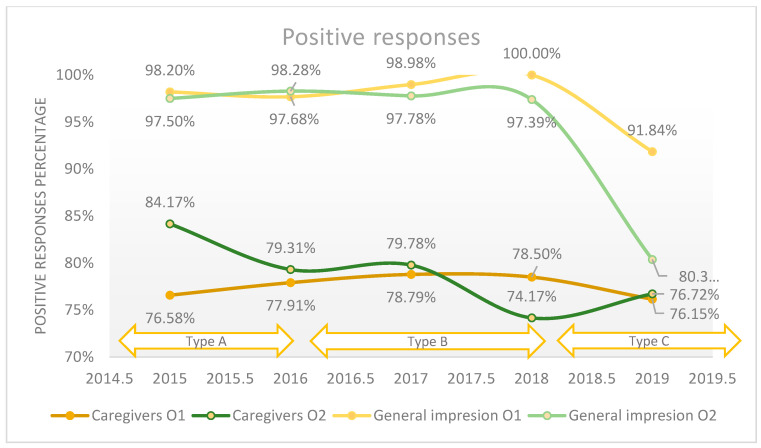
The positive responses regarding the quality of the medical care provided by caregivers in relation with patient’s overall opinion (general impression) reported annually.

**Figure 4 healthcare-11-02878-f004:**
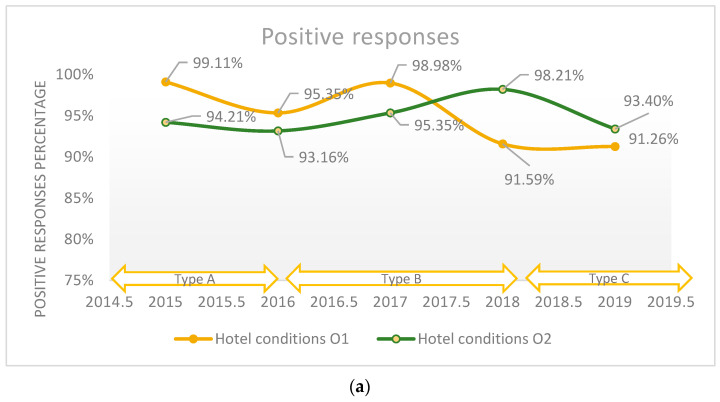
(**a**) The positive responses regarding the conditions of hospital hotel services reported annually. (**b**) The positive responses in Orthopedics 1 department regarding the conditions of hospital hotel services in relation with patient’s overall opinion (general impression) reported annually. (**c**) The positive responses in Orthopedics 2 department regarding the conditions of hospital hotel services in relation with patient’s overall opinion (general impression) reported annually.

**Figure 5 healthcare-11-02878-f005:**
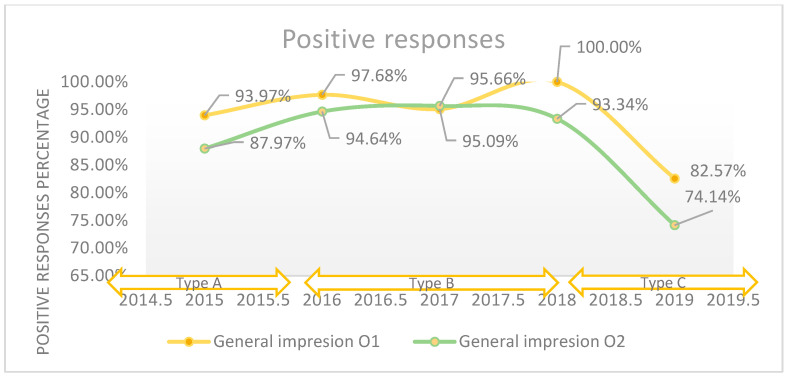
The positive responses regarding the general impression about the hospital reported annually.

**Table 1 healthcare-11-02878-t001:** Patient satisfaction survey variants used in 2015–2019.

Section/ Domain	No.	Question	Questionnaire		
			A	B	C
No. of Questions			30	31	37
Demographic data	Q1	You are: □ female □ male	X	X	X *
	Q2	Residence: □ rural □ urban	X	X	X *
	Q3	Your age: ........years old	X	X	X *
	Q4	Education: □ Higher education □ High school diploma □ 8th class/grade □ 4th class/grade	X	X	X *
Admission	Q5A, Q5B	You were admitted to the ward of...................................	X	X	-
	Q6A, Q6B	Upon admission, you were accompanied, from the Admissions Office to ward, by: □ Health personnel □ Family/friends □ I was not accompanied by anyone	X	X	-
	Q7A, Q7B	If you have been hospitalized through the Emergency department, the attitude of the staff at the Emergency Reception Room: □ Unsatisfactory □ Good □ Very good	X	X	-
	Q8A, Q8B	Specify the time elapsed from admission (through the Admissions Office—not through Emergency) until you arrived at the salon: □ Less than an hour □ An hour □ In 2 h □ In 3 h □ Over 3 h	X	X	-
	Q9A, Q9B	Give a mark (from 1 to 5) to the guard staff who dealt with you (please circle the corresponding number, 1—unsatisfactory....5—very good). □ 1 □ 2 □ 3 □ 4 □5	X	X	-
	Q10A, Q10B	** Have you been admitted to this hospital before? □ Yes □ No If the answer is YES, you returned to this hospital because: □ The medical staff is kind □ I was referred by the family doctor or specialist □ It is close to my home (it is within reach) □ I have no other options	X	X	-
	Q5C	How you came to be admitted to our hospital: □ You presented yourself directly to the Emergency department□ You had a referral from your family doctor □ You had a referral from the outpatient doctor □ You came by ambulance □ Other situation □ I do not answer	-	-	X
	Q6C	From the Admissions Office to the salon, were you accompanied by medical personnel? □ Yes □ No □ Do not answer	-	-	X
	Q7C	Were you accompanied by family members from the Admissions Office to the salon? □ Yes □ No □ Do not answer	-	-	X
	Q8C	** Have you been admitted to this hospital before? □ Yes □ No □ Do not answer	-	-	X
	Q9C	If you needed a medical service, would you return here? (1—certainly no...5-certainly yes) □ 1 □ 2 □ 3 □ 4 □ 5 □ I do not answer	-	-	X
Hotel conditions	Q11A, Q11B	The food in the hospital is: □ Good □ Very good □ Not tasty □ Too little	X	X	-
	Q12A, Q12B	You received the food on time: □ Yes □ No □ I do not know	X	X	-
	Q13A, Q13B	** Specify how many times a day your salon/room is cleaned: □ Once a day □ 2 times a day □ 3 times a day □ Several times a day	X	X	-
	Q14A, Q14B	** The cleanliness of your salon is: □ Non-existent □ Good □ Very good	X	X	-
	Q15A, Q15B	** Are you satisfied with the hotel conditions (accommodation)? □ Very satisfied □ Satisfied □ Unsatisfied	X	X	-
	Q16B	What do you think about the quality of the linen? □ Non-existent □ Good □ Very good	-	X	-
	Q10C	The quality of food in the hospital is: □ 1 □ 2 □ 3 □ 4 □5 □ I do not answer	-	-	X
	Q11C	What do you think about the quality of sanitary groups? □ 1 □ 2 □ 3 □ 4 □5 □ I do not answer	-	-	X
	Q12C	** Specify how many times a day your salon/room is cleaned: □ Once a day □ 2 times a day □ As many times as necessary □ Do not answer	-	-	X
	Q13C	** The cleanliness of your salon is: □ 1 □ 2 □ 3 □ 4 □ 5 □ I do not answer	-	-	X
	Q14C	** Are you satisfied with the accommodation conditions—salon (equipment, facilities)? □ 1 □ 2 □ 3 □ 4 □ 5 □ I do not answer	-	-	X
Quality of medical care	Q16A, Q17B	The time allotted by the salon doctor for your consultation: □ Unsatisfactory □ Good □ Very good	X	X	-
	Q17A, Q18B	** The quality of medical care provided by: Your doctor: □ Unsatisfactory □ Good □ Very good Nurses: □ Unsatisfactory □ Good □ Very good Caregivers: □ Unsatisfactory □ Good □ Very good	X	X	-
	Q18A, Q19B	Were you satisfied with the care provided? During the day: □ Yes □ No During the night: □ Yes □ No Saturday, Sunday, and public holidays: □ Yes □ No	X	X	-
	Q19A, Q20B	How do you rate the contact with the hospital staff? □ Agreeable □ Cold/impersonal □ Warm/close □ Unpleasant	X	X	-
	Q15C	** The quality of medical care provided by: Your doctor: □ 1 □ 2 □ 3 □ 4 □ 5 □ I do not answer Nurses: □ 1 □ 2 □ 3 □ 4 □ 5 □ I do not answer Caregivers: □ 1 □ 2 □ 3 □ 4 □ 5 □ I do not answer (1—unsatisfactory....5—very good)	-	-	X
	Q16C	How do you rate the attitude of the hospital staff? (1—unsatisfactory....5—very good) □ 1 □ 2 □ 3 □ 4 □ 5 □ I do not answer	-	-	X
	Q17C	Did you have surgery during your hospitalization? □ Yes □ No	-	-	X
	Q18C	How do you rate the post-operative care and medical care provided in the Intensive Care Unit (if applicable)? □ 1 □ 2 □ 3 □ 4 □ 5 □ Not applicable/no answer	-	-	X
	Q19C	Were you satisfied with the spiritual care in the hospital? □ Yes □ No □ Do not answer	-	-	X
Patient’s safety and rights	Q20A, Q21B	** During the hospitalization you bought medicines: □ Yes □ No	X	X	-
	Q21A, Q22B	** Can you name a medicine that was administered to you in the hospital? □ Yes □ No □ Specify the name of the medicine…………………………..	X	X	-
	Q22A, Q23B	Do you know at least one adverse effect of the administered drug or risk of the procedure performed? □ Yes □ No □ Specify.....................................................	X	X	-
	Q23A, Q24B	Was the administration of oral medications (tablets) done by the nurse? □ Yes, always □ Yes, sometimes □ No, never	X	X	-
	Q24A, Q25B	From whom did you learn your diagnosis (disease)? □ The hospital doctor □ Salon assistant □ I do not know my diagnosis	X	X	-
	Q25A, Q26B	Do you know what kind of tests were done in the hospital? □ Yes □ No □ I was not tested	X	X	-
	Q26A, Q27B	** Have you been informed about your rights in the hospital? □ Yes, at the admission office □ Yes, only verbally □ No	X	X	-
	Q27A, Q28B	Did you have surgery during your hospitalization? □ Yes □ No Have you talked to your doctor about the surgery you had and its risks? □ Yes, everything was explained to me step by step □ I did not receive any explanation	X	X	-
	Q28A, Q29B	** Is the visiting schedule observed on the ward where you were admitted? □ Yes □ No	X	X	-
	Q20C	Were you accompanied by designated staff when moving around the hospital (for explorations)? □ Yes □ No □ Do not answer	-	-	X
	Q21C	Do you know the identity of the medical personnel involved in providing medical services? □ Yes □ No □ Do not answer	-	-	X
	Q22C	** During the hospitalization, you bought medicines? □ Yes □ No □ Do not answer	-	-	X
	Q23C	** Can you name a medicine that was administered to you in the hospital? □ Yes □ No □ Do not answer	-	-	X
	Q24C	Have you been informed about the adverse effects of the drugs administered in the hospital? □ Yes □ No □ Do not answer	-	-	X
	Q25C	Have the vials administered been opened in front of you? □ Yes □ No □ Do not answer	-	-	X
	Q26C	Do medical personnel use disposable gloves in every contact with you? □ Yes □ No □ Do not answer	-	-	X
	Q27C	** Have you been informed about your rights and obligations in the hospital? □ Yes, at the admission office □ Yes, only verbally □ No □ Do not answer	-	-	X
	Q28C	Have you been informed about the way to submit suggestions and complaints? □ Yes □ No □ Do not answer	-	-	X
	Q29C	Have you been informed of the estimated date of discharge? □ Yes □ No □ Do not answer	-	-	X
	Q30C	Have you been informed about the risk of falling? □ Yes □ No □ Do not answer	-	-	X
	Q31C	Were you informed about your diagnosis? □ Yes □ No □ Do not answer	-	-	X
	Q32C	Did you receive information about how the disease/illness will evolve and the therapeutic plan followed? □ Yes □ No □ Do not answer	-	-	X
	Q33C	During your hospitalization, did you reward any medical staff (doctor, assistant, nurse, carer, stretcher bearer, etc.) with money or gifts? □ Yes □ No □ Do not answer	-	-	X
	Q34C	If the answer was YES to the previous question, please specify the professional category of the medical staff: □ Doctor □ Caregivers □ Others □ Nurse □ Stretcher bearer □ Not applicable/no answer	-	-	X
	Q35C	** Is the visiting schedule observed on the ward where you were admitted? □ Yes □ No □ Do not answer	-	-	X
Overall satisfaction	Q29A, Q30B	Your general impression of the hospital? □ Very satisfied □ Satisfied □ Unsatisfied	X	X	-
	Q36C	Your general impression of the hospital? □ 1 □ 2 □ 3 □ 4 □ 5 □ I do not answer	-	-	X
Observations/suggestions	Q30A, Q31B, Q37C		X	X	X

Note: * Observations: In Questionnaire C, one answer option was added to all questions: □ I do not answer; ** = similar questions.

**Table 2 healthcare-11-02878-t002:** Questionnaire types.

Characteristic			
Questionnaire	Type A	Type B	Type C
Period	2015–2016	2017–2018	2019
No. of questions	30	31	37
Quality assessment with Likert scale	Three-point scale and five-point scale		Five-point scale
	(3–5 values)		(5 values)
Correspondence of scales	Very good -> 5–4		5—Very good
			4—Medium good
	Good -> 3		3—Good
			2—Satisfactory
	Unsatisfactory -> 1–2		1—Unsatisfactory

**Table 3 healthcare-11-02878-t003:** Sample characteristics for each department.

Characteristic	N (%)			*p*-Value *
Department	Orthopedics 1 (O1)	Orthopedics 2 (O2)	Total subjects	
Study sample	520 (47.2%)	581 (52.8%)	1101 (100%)	
Sex				0.042
Male	231 (44.4%)	295 (50.7%)	522 (47.8%)	
Female	289 (55.6%)	286 (49.2%)	575 (52.2%)	
Residence				0.030
Urban	238 (45.8%)	286 (49.2%)	445 (45.93%)	
Rural	228 (43.98%)	217 (37.4%)	524 (54.07%)	
Declined to answer	54 (10.4%)	78 (13.4%)	132 (11.98%)	
Education				0.046
Higher education	204 (39.2%)	205 (35.3%)	409 (38.26%)	
High school diploma	99 (19%)	95 (16.4%)	194 (18.14%)	
8th class/grade	156 (30%)	201 (34.6%)	357 (33.40%)	
4th class/grade	46 (8.8%)	63 (10.8%)	109 (10.20%)	
Declined to answer	15 (2.9%)	17 (2.9%)	32 (2.91%)	

* Pearson’s chi-squared test applied for each department.

**Table 4 healthcare-11-02878-t004:** The evaluation of the Likert scale internal consistency.

Questionaire	Type A		Type B		Type C
	2015	2016	2017	2018	2019
Cronbach’s alpha value *	0.596197		0.246734		0.706937

* Internal consistency of the Likert scale used.

## Data Availability

Data sets analyzed or generated during the study are available by requesting them from the authors at the email address: arcmaeoffice@yahoo.com.

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
