# Peer review of "Comparative Analysis of Patient Satisfaction Surveys—A Crucial Role in Raising the Standard of Healthcare Services"

_healthcare, 2023, doi:10.3390/healthcare11212878_

Round 1

Reviewer 1 Report

Comments and Suggestions for Authors

Dear authors,

Many thanks for submitting your valuable work to the journal. I have made specific comments on your work that are presented in the below-attached file.

Best regards

Reviewer 2 Report

Comments and Suggestions for Authors

The manuscript is coherent and shows interesting data about the quality of medical care perceived by patients, which can be a valuable tool to establish healthcare satisfaction.
However, in my opinion, some suggestions should be addressed by the authors before recommending its publication.
The authors compare the data obtained from two wards to an orthopedic service. The analysis of the demographic data between respondents from both wards shows significant statistical differences, so it is suggested that the authors add a paragraph in the discussion discussing whether or not this difference may influence their results. 

The authors propose two main objectives of their study, to evaluate the level of satisfaction with the quality of the health service and to compare 3 tools to know the quality of service. In the case of the analysis of the tools they did not perform a validation analysis of each device that would have yielded data about each construct.

The graphic could be improve. 

Round 2

Reviewer 1 Report

Comments and Suggestions for Authors

Dear authors,

Many thanks for your answer to my previous comments. The manuscript has now been significantly improved and revised based on my evaluation. However, I suggest providing a sub-section named "Ethics" in the "Material and Methods" section, according to one of my previous comments that have not been addressed.

Best regards

Author Response

Dear Reviewer,

Thank you very much for your feedback. We adapted the paper as suggested. We added a subsection named "Ethics" in the "Material and Methods" section.

We look forward working together on other papers as well. Thank you for your support and help.